# A CBCT Study of Labial Alveolar Bone Thickness in the Maxillary Anterior Region in a Teaching Hospital Population in the Eastern Province of Saudi Arabia

**DOI:** 10.3390/biomedicines11061571

**Published:** 2023-05-29

**Authors:** Abdulmajeed A. Aljabr, Khalid Almas, Faisal E. Aljofi, Abdullah A. Aljabr, Bader Alzaben, Sarah Alqanas

**Affiliations:** 1Fellowship in Periodontics Program, College of Dentistry, Imam Abdulrahman Bin Faisal University, P.O. Box 1982, Dammam 31441, Saudi Arabia; 2210600051@iau.edu.sa; 2Department of Preventive Dental Sciences, College of Dentistry, Imam Abdulrahman Bin Faisal University, P.O. Box 1982, Dammam 31441, Saudi Arabia; kalmas@iau.edu.sa (K.A.); fealjofi@iau.edu.sa (F.E.A.); 3Department of Medical Education, College of Dentistry, Majmaah University, P.O. Box 1712, Al Majma’ah 15341, Saudi Arabia; aa.jabr@mu.edu.sa; 4College of Dentistry, Imam Abdulrahman Bin Faisal University, P.O. Box 1982, Dammam 31441, Saudi Arabia; 2160002964@iau.edu.sa

**Keywords:** labial bone, buccal plate, maxillary anterior teeth, alveolar bone thickness, CBCT

## Abstract

Background and Objectives: Labial alveolar bone thickness in the maxillary anterior region is the key factor in the placement of implants. Differences in the thickness of the bone are reported among different ethnic groups. Thus, the present study was aimed at assessing labial alveolar bone thickness in the maxillary anterior region in the population of the eastern province of Saudi Arabia. Materials and Methods: The six anterior teeth in each of the 186 CBCT sagittal images were analyzed at three points: Point A from the facial plate at the level of the bone crest to the coronal root third, Point B to the mid-root surface, and Point C to the apical third. Crest height (Point D) was measured as the distance from the CEJ to the alveolar bone crest. The analysis was done using SPSS version 20. A *p*-value of <0.05 was considered statistically significant. Results: The bone thickness at any given point (Point A, Point B, or Point C) was less than the preferred bone thickness of 2 mm in all six teeth. The thickness was minimum at Point B, maximum at Point C, and intermediate at Point A. This difference was found to be statistically significant (*p*-value < 0.05). The crestal height (Point D) was less than 3 mm, and it was not statistically significant. Comparison of bone thickness on the right and left sides for any given point was not statistically significant except at Point A in the central incisor, where it was statistically significant (*p*-value = 0.035). Gender comparison of bone thickness showed no difference at Point A; however, at Points B and C, it showed statistical significance (*p*-value < 0.05). Conclusions: The alveolar bone thickness being <2 mm in the labial anterior region warns of the importance of the proper assessment of bone during implant placement to have a predictable outcome.

## 1. Introduction

Alveolar bone thickness in the anterior or posterior region plays a vital role in the treatment planning for a successful implant [1]. Alveolar bone height, width, morphology, the density of alveolar bone, and facial or buccal bone thickness in the proposed implant site are important parameters [2]. Facial bone thickness in the maxillary anterior region is of aesthetic and functional importance [3]. Extraction and healing in maxillary anterior teeth sockets result in the loss of cortical bone, compromising more bone on the facial side than on the palatal side [4]. Often, this results in thinner bone on the facial aspect or the occurrence of fenestration and dehiscence. The anterior region, given the aesthetic concerns, requires immediate implant placement and immediate restoration [5]. Unfortunately, in this region, the required bony thickness failed to meet this demand. Furthermore, the presence of excessive gingival display and the shorter upper lip complicates the treatment of this aesthetic corridor [6]. There is an established bidirectional relationship between buccal bone thickness and crestal labial soft tissue. Thus, the anatomical importance of this region has drawn the attention of clinicians and researchers to conduct studies to assess facial bony thickness [7,8,9,10,11]. In the Kingdom of Saudi Arabia, several studies are being conducted in different parts of the country [12,13,14,15,16,17,18]. The outcome of this study showed variation in the results among the gender, age group, right and left side, and for different teeth. Furthermore, there had been no studies conducted in the Eastern province of Saudi Arabia. 

Thus, the present study was aimed at assessing the labial alveolar bone thickness of the maxillary anterior region as a primary objective. The secondary objectives of the study were a comparison of the labial bone thickness among male and female patients, right and left sides, and selected age groups. 

## 2. Materials and Methods

### 2.1. Study Design

Sample size—For the study, CBCT images of patients who attended a university dental clinic between June 2021 and December 2021 were selected. A total of 506 CBCT scans were obtained from the database at the Radiology Department, University Dental Hospital, Imam Abdulrahman bin Faisal University, Kingdom of Saudi Arabia.

Sample size calculation—Assuming mean mid-buccal bone thickness was 0.83 and apical bone thickness was 1.06, standard deviation 0.49 with 80% power, 5% level of significance, and 95% confidence level, the required sample size was 186 buccal bone CBCT images of maxillary teeth. The World Health Organization Sample Size calculator (Lun & Chiam, National University of Singapore, software version) was used for the sample size calculation [18].

Inclusion and exclusion criteria:

Inclusion criteria:a.Saudi patients, age ranged between 18 to 65 years.b.Non-smokers. c.Absence of any systemic disease. d.Periodontal health—absence of any signs of periodontal disease.e.Presence of normal occlusion.f.The presence of all six maxillary anterior teeth (including canine). 

Exclusion criteria: a.Poor image quality of CBCT scan.b.Images from patients with systemic or pathological dentoalveolar conditions (e.g., cyst) that might cause abnormal bone remodeling.c.Any history or current periapical lesion. d.Presence of inflammatory processes at the apical level.e.Patient with a previous history of road traffic accidents (RTA). f.Cancer subjects. g.History of radiation or chemotherapy.h.Osteoporosis conditions.i.Tooth malalignment. 

All the patient records were searched according to these inclusion and exclusion criteria, and the final required number of CBCT scans was selected. 

### 2.2. Radiographic Image Analysis: Detail of CBCT

All the CBCT images selected for the study were exposed from the same machine. (Orthophos (Sirona Dental Systems, Bensheim, Germany). Imaging parameters used were 85 kV, 6 mA, 14.1 s exposure time, 0.2 mm voxel size, and 80 × 40 mm field of view. Imaging parameters were determined according to the “as low as reasonably achievable” (ALARA) principle. The images were constructed and analyzed using Horos 3.0 software (Horos Project, Annapolis, MD, USA). 

### 2.3. Scan Measurements

Measurements on the CBCT scans were analyzed by two independently trained residents. The right-side canine (RCA), right-side lateral incisor (RL), right-side central incisor (RC), left-side canine (LCA), left-side lateral incisor (LL), and left-side central incisor (LC) were included in the analysis. Prior to the measurements, two observers (B.A. and S.A.) were calibrated with the evaluation of 20 CBCT images. The weighted mean kappa score was 0.92. All measurements were performed twice by one observer, and the averages were submitted to statistical analysis. Secondly, only three CBCT images were measured at one time, and after every three CBCT measurements, a break was taken to avoid eye fatigue of the observer. Repeated measurements of reliability between investigators were assessed to measure their degree of agreement. For this, the CBCT images were examined one week apart (every 5th CBCT was re-examined by the examiners to rule out intra- and inter-examiner variability). Intraclass correlation coefficients were calculated to assess outcome reproducibility and consistency between all repeated measures.

### 2.4. Data Collection

#### Alveolar Bone Thickness Measurement

At each tooth, the facial plate thickness of the alveolar bone was measured from a sagittal CBCT image view of the tooth root. The sagittal section was made at the middle of each tooth by applying the cursor in the midline that bisects the tooth into equal halves. Reference points were used to measure alveolar bone thicknesses at three locations using a digital caliper: Point A from the facial plate at the level of the bone crest to the coronal root third, Point B to the mid root surface, and Point C to the apical third. All measurements were taken in millimeters (mm). To set fixed reference points for each tooth in the sagittal view, the cursor was placed at the tooth’s midline, and in the sagittal view, the tooth root was divided equally into cervical, middle, and apical thirds. Reference points were set at the midpoint of each third, while the cementoenamel junction (CEJ) was set as a fixed reference point for measuring crest height. Crest height (Point D) was measured as the distance from the CEJ to the alveolar bone crest. This was done by using the same sagittal view as that used for measuring thicknesses and the same digital caliper as mentioned above. The built-in digital caliper was used for direct bone measurements on the CBCT images. All images were viewed on the same monitor and under the same lighting conditions (Figure 1). 

### 2.5. Data Analysis

All the analysis was done using SPSS version 20. A *p*-value of <0.05 was considered statistically significant. Buccal bone thickness was measured with descriptive analysis using mean ± standard deviation (SD) and median (min-max). Comparison of variables with gender and age groups was done using an independent sample *t*-test. A comparison of variables between the right and the left side was done using paired *t*-test. 

## 3. Results

To assess the outcome reproducibility and consistency between all repeated measures, intraclass correlation coefficients were calculated which ranged from 0.89–0.95.

### 3.1. Demographic Data

Among the 186 CBCT scans examined, there were 102 males (54.83%) and 84 (45.17%) were females. For convenience, age groups were divided into those below and above 30 years of age. The groups consisted of 114 (61.29%) (below 30 years of age) and 72 (38.71%) (above 30 years of age). 

### 3.2. Alveolar Bone Thickness 

The bone thickness at any given point (Point A, Point B, and Point C) was less than the preferred thickness of bone of 2 mm in all the six teeth measured. Moreover, only in the lateral incisor were 13% of teeth between 1.5 to 2 mm (Table 1). 

### 3.3. Comparison of Bone Thickness at Different Points

In the six anterior teeth measurements taken at the three points A, B, and C, it was found that the thickness was least at Point B (0.76 mm to 0.85 mm) and the greatest at Point C (1.34 to 1.50 mm). At Point A, the bony thickness was intermediate (0.97 to 1.13 mm). This difference was found to be statistically significant (*p*-value <0.05). (Figure 1). However, a comparison among the teeth was not statistically significant (*p*-value ≥ 0.05) (Table 2).

### 3.4. Comparison of Bone Thickness—Right and Left Sides

The comparison of the bone thickness on the right and left sides at any given point (Point A, B, or C) was not statistically significant between the six teeth measured (canine, lateral, and central incisor), except at Point A in the central incisor, where it was statistically significant (*p*-value = 0.035) (Table 2, Figure 2).

### 3.5. Comparison of Bone Thickness by Gender

The comparison of bone thickness for all six teeth between males and females showed no statistical significance at Point A (*p*-value ≥ 0.05). However, for Point B, in the right canine, right central incisor, and left canine, and for Point C, in the right lateral incisor and left canine, there was a statistically significant difference (*p*-value < 0.05) (Table 3).

### 3.6. Comparison of Bone Thickness among Different Age Groups

The comparison of the bone thickness at different given points compared between the two age groups (below and above 30 years) showed no statistically significant difference (*p*-value ≥ 0.05) (Figure 3).

### 3.7. Alveolar Bone Height 

The mean alveolar bone height was measured as the distance from the cementoenamel junction (CEJ) to the alveolar crest (Point D) in the sagittal view of the CBCT. All six anterior teeth examined showed that the distance was less than 3 mm, and it was not statistically significant (*p*-value ≥ 0.05) (Table 4). Comparing the alveolar bone height in males and females showed a statistically significant difference (*p*-value ≤ 0.05) (Figure 4).

## 4. Discussion

The clinical implication of facial bone thickness and its variation in different populations and races prompted us to conduct this study. To date, there have been seven studies which have been conducted and have reported facial bone thickness in the Kingdom of Saudi Arabia. Few studies have considered all the six anterior teeth, and some of them considered only a single tooth, such as the canine [11,12,13,14,15,16,17]. Although there was some similarity in the results among these studies in the data obtained (bone thickness in a particular tooth or alveolar crestal bone height), still, there were differences, and varied findings have been observed. Furthermore, there was no study conducted in the Eastern province of the Kingdom of Saudi Arabia. Thus, the outcomes of this study are expected to deliver new information about facial bone thickness among the studied population. 

The primary outcome of the study was to assess the facial bone thickness in relation to six maxillary anterior teeth. In all measured points, we have found the bone thickness was less than 2 mm and, in some areas, it was less than 1 mm. These findings of our study were in accordance with the result of Sheerah et al. [15] and Othman et al. [14]. In both of these studies, they measured the mean alveolar bone thickness and reported that, in most cases, the bone thickness was less than 1.5 mm. In a study carried out by Almahdi et al. [11], it was reported that the mean thicknesses of the facial bone at three levels were 0.83, 0.84, and 1.06 mm (at the coronal, middle, and apical levels). AlTarawneh et al. [19] found similar results to Almahdi et al., with the mean values of bone thickness at coronal, middle, and apical thirds of the labial side for central incisor roots being 0.73, 0.69, and 0.60 mm; for lateral incisors, 0.70, 0.61, and 0.49 mm; and for canines, 0.74, 0.53, and 0.40 mm. Our results were in agreement with AlTarawneh’s study regarding the apical and middle point buccal bone thickness. This could be due to the fact that both the populations were the same in terms of ethnic background and age. The preferred buccal bone thickness to prevent dehiscence or post-surgical recession was ≥2mm [20,21]. A thinner buccal bone and the presence of an undercut may increase the risk of fenestration, cortical bone perforation, and even recession, during or after the implant procedure [22]. A systematic review of immediate implant placement with bone grafting in preserving the buccal plate thickness revealed that immediate implants with bone grafting had superior soft-tissue stability and preserved horizontal-ridge dimension and buccal plate thickness when compared to no grafting. The use of a barrier alone significantly decreased buccal plate resorption and the remaining defects around the implants, and the use of both bone graft and membrane aided in soft-tissue preservation [23]. Thus, bone grafting and other regenerative procedures may be required to avoid complications such as facial gingival recession, exposure of implant threads, and peri-implant diseases. Even restoring this space with immediate implant placement is to be done cautiously [24].

In this study, interestingly, in all six anterior teeth, at the midpoint or Point B, there was less thickness of the bone compared to coronal (Point A) and apical (Point C). This is an important finding because negligence in considering this finding may lead to mid-socket fenestration, which often goes unnoticed. Othman et al. [14] reported similar findings, with the difference that in the central incisor they found significantly greater midpoint buccal thickness compared to the lateral incisor and canines. Our study results were not in agreement with the studies of Nahás-Scocate et al. [25] and Linjawie et al. [12], who reported alveolar bone thickness to be greater at the apex level followed by the mid-root and the alveolar crest levels. In our study, mid-buccal readings were lesser than for the alveolar crest level. Buccal prominence of the canine and concavity of the maxilla in the lateral incisor region may have led to the thinner bone in this area [22,26].

Variation in our study also could be due to the sample-size difference, variability in the root design (shape), and angulation of the root and inclination of the maxillary incisors in the arch. Studies have reported significant changes in the thickness of the maxillary alveolar bone following alterations in the inclination of the maxillary incisors [27,28,29,30]. In the present study, comparison of the left side to the right side showed no statistically significant difference except in one point (Point A) in the central incisor region. The studies of Othman et al. [14], Sherrah et al. [15], and Alsaffar et al. [16] were in accordance with the present study. However, in contrast, one study reported differences in some points, only on the left side, to be significantly different in some measurements of buccal bone thickness according to age and gender [30]. Farahamnd et al. [31] concluded that there were some recognizable differences in the facial bone thickness between the left and the right side. Another study reported no significant difference between right and left in central and lateral incisors; however, the canine showed a significant difference [32]. Hassan et al. found differences only in the apical level between the right and left sides in both females and males [33]. Thus, although there are a few regional studies in accord with the results of our study, other studies’ outcomes were contradictory to our findings. The reason for the variation may be ethnic and racial differences, the methods used, or technical aspects of CBCT. 

Gender variation in the buccal bone thickness was evaluated to see if there is any probability of variation in the bone thickness. We have found differences in bone thickness between males and females, although not in all the points measured. Nowzari et al. [20], Morad et al. [32], Almahdi et al. [11], Sherrah et al. [15], Zhang et al. [22], Üner et al. [34], and Hassan et al. [33] found similar results, with males being reported to have thicker bone compared to females. Interestingly, Alsaffar et al. [16] found females to have a lesser bone thickness in the apical and middle-third of the root than males, while males had overall thinner alveolar bone than females. However, Othman et al. [14], Lim et al. [35], and Ghassemian et al. [36] did not find any difference between the genders. Ohiomoba et al. [37], in their study of the anterior maxilla, reported female subjects to have significantly denser bone compared with male subjects; however, gender was not significantly associated with bone thickness. The difference in the results among these various studies may be due to sampling size, male-to-female ratio, and parameters taken into consideration (few studies have taken mean bone thickness, and other studies have taken bone thickness at each point). 

The relation between the age and the thickness of the alveolar bone in the maxillary anterior teeth has been explored in the present study. We found no difference in the age groups compared (below 30 years and above 30 years). The study results of Nowzari et al. [20] and Almahdi et al. [11] were similar to our study results regarding age. However, Fuentes et al. [30] found that only on the left side of the maxillary anterior teeth were there significant differences in some measurements of buccal bone thickness observed according to age and gender. Januário et al. [38] found differences in the younger and older age groups. Another study by Gakonyo et al. [39] reported decreased bone thickness as age advances. Ohiomoba et al. [37] reported body mass index and age are positively associated with bone thickness and density. Thus, as it is seen with gender, a few studies have reported differences in bone thickness between younger and older age groups, and other studies have concluded that there are no differences. The sample size differences and parameters chosen in the CBCT study may be the reason for the differences seen among these studies.

In the present study, Point D, the distance from the CEJ to the alveolar crest, was in the range of 1.45 ± 40 to 1.56 ± 43. Typical alveolar crest to CEJ distance is reported to be 2 mm apical to the CEJ [40]. The Zhou et al. [41] findings regarding the alveolar crest height were very close to our study (1.80 ± 0.56 mm). Januário et al. [38] reported about 1.6 and 3 mm, which are also close to the findings of our study. However, other studies have reported a high variation in CEJ–bone crest (0.8 to 7.2 mm) [28]. Studies done in the Kingdom of Saudi Arabia have reported slightly higher ranges of 2.66 ± 0.88 to 3.06 ± 1.09 mm [12], 2.84 ± 2.01 to 5.68 ± 4.91 [16], and 2.09 ± 0.66 to 2.74 ± 0.81 [15]. If the distance between the CEJ to the alveolar crest is more than 3 mm, there may be more chance of the occurrence of dehiscence in the implant site. The differences seen in the distance from CEJ to the crest of the alveolar bone may be due to ethnic and racial variations. Technically, the types of CBCT machines, digital calipers, and software used may also have had an impact on this measurement.

There are limitations of this study. Bone thickness depends upon the inclination of the tooth in the arch, and the tooth form, root form, and other morphological features. We have not included in our study, in future studies, if these factors are included, it may help to delineate the findings, which may help in making better correlations among the teeth studied. Although we had a limited study sample of Saudi citizens, ethnic population differences were not taken into consideration, and this probably would help yield variable results in future studies.

## 5. Conclusions

Facial bone thickness in the selected six anterior teeth was found to be less than the preferred bone thickness of >2 mm, with the mid-buccal point being thinner than the apical and coronal bone thickness. There was no difference in the buccal bone thickness between the right and left sides and the different age groups. However, differences in bone thickness were seen in a few teeth between the male and female subjects. Furthermore, the so-called jumping distance (that is, the distance between the implant and bone), if it is more than 2 mm, may result in the collapse of the buccal bone, resulting in fenestration or dehiscence. In the present study, in relation to the lateral incisor, the buccal plate thickness was much greater than that of the canine and the central incisor, where the placement of the implant could be closer to the plate.

Thus, one has to be cautious about implant placement in the anterior maxilla, especially any immediate implant placement in the studied population.

## Figures and Tables

**Figure 1 biomedicines-11-01571-f001:**
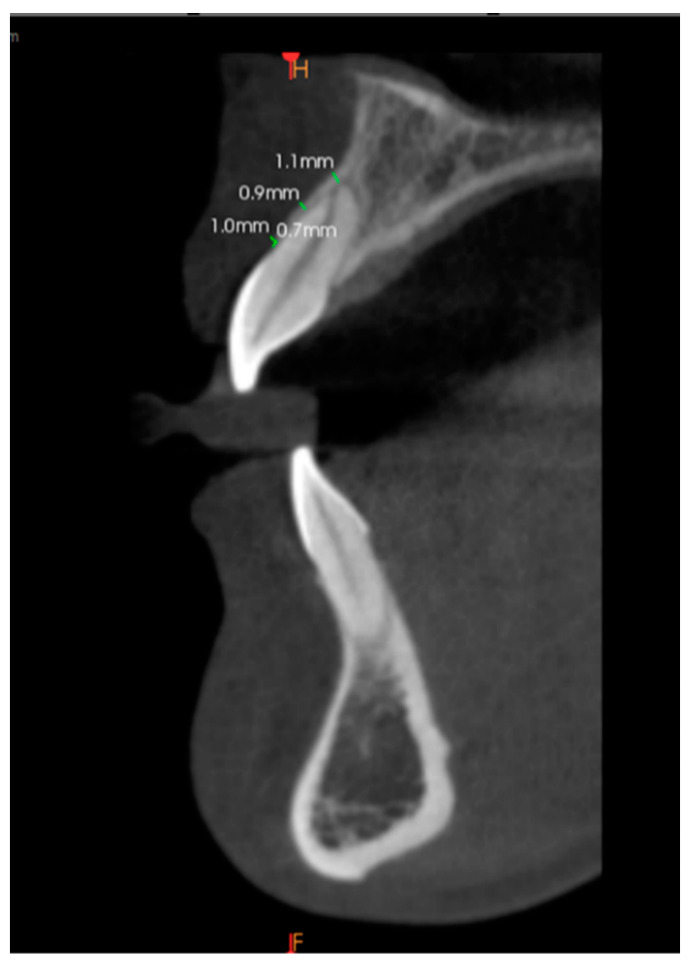
Labial bone thickness measurement at Points A, B, and C.

**Figure 2 biomedicines-11-01571-f002:**
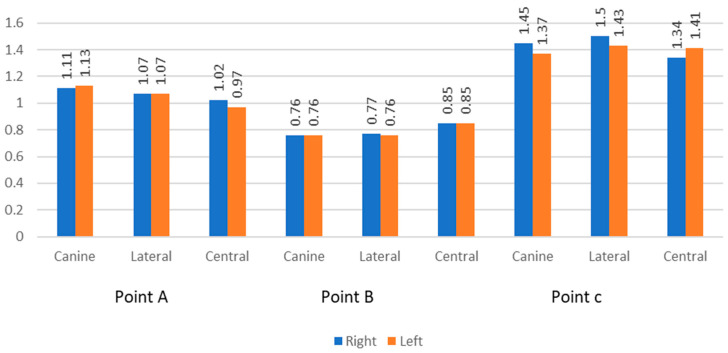
Comparison of labial bone thickness at Point A, Point B, and Point C in six maxillary anterior teeth.

**Figure 3 biomedicines-11-01571-f003:**
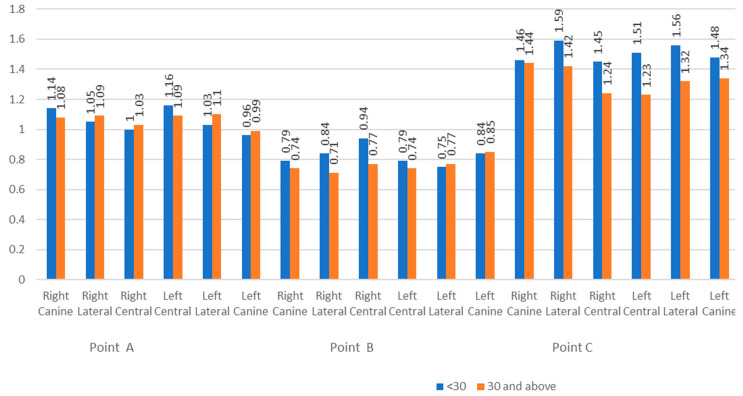
Comparison of the labial bone thickness between the age groups.

**Figure 4 biomedicines-11-01571-f004:**
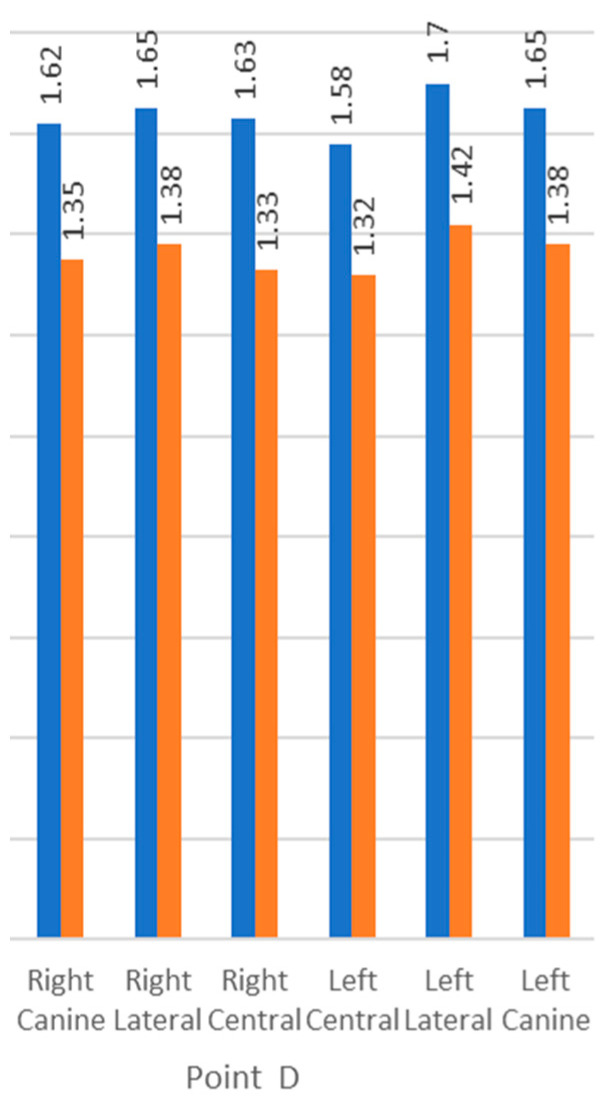
Comparison of Point D (distance from CEJ to the crest of the alveolar bone) among males (blue) and females (orange).

**Table 1 biomedicines-11-01571-t001:** The percentage and frequency distribution of the labial bone thicknesses.

	Canine (*n*%)	Lateral Incisor (*n*%)	Central Incisor (*n*%)
<1.5	100	87	100
1.5 TO 2	Nil	13	Nil
>2	Nil	Nil	Nil

**Table 2 biomedicines-11-01571-t002:** Comparison of the labial bone thickness among maxillary anterior teeth.

Point	Tooth	Right		Left		*p*-Value
		Mean	SD ± mm	Mean	SD ± mm	
A	Canine	1.11	0.30	1.13	0.34	0.556
	Lateral	1.07	0.27	1.07	0.31	>0.99
	Central	1.02	0.20	0.97	0.20	0.035
B	Canine	0.76	0.25	0.76	0.27	0.944
	Lateral	0.77	0.41	0.76	0.33	0.823
	Central	0.85	0.28	0.85	0.26	0.896
C	Canine	1.45	0.47	1.37	0.48	0.111
	Lateral	1.50	0.73	1.43	0.73	0.194
	Central	1.34	0.49	1.41	0.52	0.254

**Table 3 biomedicines-11-01571-t003:** Gender comparison of labial bone thickness.

			Sex				*p*-Value
			Male		Female		
			Mean	SD	Mean	SD	
**Point**	A	Right Canine	1.10	0.29	1.12	0.31	0.853
		Right Lateral	1.08	0.26	1.06	0.29	0.76
		Right Central	1.03	0.20	1.01	0.21	0.68
		Left Central		0.22	0.97	0.18	0.944
		Left Lateral		0.28	1.02	0.35	0.285
		Left Canine		0.30	1.11	0.38	0.713
	B	Right Canine	0.87	0.26	0.66	0.20	0.003
		Right Lateral	0.83	0.50	0.71	0.29	0.285
		Right Central	0.94	0.31	0.76	0.22	0.024
		Left Central	0.89	0.29	0.80	0.22	0.217
		Left Lateral	0.79	0.35	0.73	0.32	0.559
		Left Canine	0.85	0.30	0.67	0.21	0.017
	C	Right Canine	1.57	0.52	1.33	0.38	0.071
		Right Lateral	1.71	0.87	1.30	0.50	0.048
		Right Central	1.44	0.58	1.24	0.37	0.138
		Left Central	1.46	0.53	1.36	0.51	0.481
		Left Lateral	1.57	0.83	1.30	0.61	0.199
		Left Canine	1.51	0.51	1.22	0.41	0.036
	D	Right Canine	1.62	0.40	1.35	0.41	0.021
		Right Lateral	1.65	0.51	1.38	0.39	0.043
		Right Central	1.63	0.42	1.33	0.38	0.013
		Left Central	1.58	0.43	1.32	0.33	0.019.
		Left Lateral	1.70	0.44	1.42	0.37	0.017
		Left Canine	1.65	0.50	1.38	0.39	0.04

**Table 4 biomedicines-11-01571-t004:** Comparison of alveolar bone crest height (Point D) (distance from CEJ to the crest of the alveolar bone).

Point	Tooth	Right		Left		*p*-Value
		Mean	SD ± mm	Mean	SD ± mm	
D	Canine	1.48	0.42	1.51	0.46	0.453
	Lateral	1.51	0.47	1.56	0.43	0.171
	Central	1.48	0.43	1.45	0.40	0.443

## Data Availability

All the supporting data are available within the article. Further data available and presented in this study are available on request from the corresponding author.

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
