# Peer review of "A CBCT Study of Labial Alveolar Bone Thickness in the Maxillary Anterior Region in a Teaching Hospital Population in the Eastern Province of Saudi Arabia"

_biomedicines, 2023, doi:10.3390/biomedicines11061571_

Round 1

Reviewer 1 Report

A well planned and a great team effort to put together this research. Congratulations.

Please do the following corrections to improve your study

1. Lines 112-115 is not clear.

2. The test mentioned in line 115 is not showing results.

3. Line 122 - Is the measurement from the outer surface of the facial plate?

(Measurements done from 122 to 124 depends on the root surface. Would it be practical since the alveolar bone plate can have variation in its position on the root surface. Since the distance from the CEJ to Crest is not constant. Please justify this with appropriate reference.)

4. Lines 231-234 - repetition of the methodology and results (not necessary to repeat). Please add references on immediate implant bone grafting studies and their view on the facial plate thickness.

5. Lines 245 - 251 - why explain it here? Since it is the main  reason for conducting this study. 

6. Significance of Point D is not discussed in the discussion section.

7. Putting sample size as a limitation may not appropriate as the sample size was methodically calculated and the Level of significance was 5%. 

8. Inclusion of tooth inclination and buccopalatal width in future studies is a good future prospect rather than limitation.

9. In conclusion instead of 2mm buccal bone the amount of distance from the implant surface to the buccal plate to fulfill the 2mm gap could be explained. As most of the immediate implant case needs grafting to avoid any collapse of the labial plate. Also inclusion of lateral incisor buccal plate thickess is vital here, placement of implant here could be closer to the plate. 

The English language in terms of grammar corrections have to be done in the entire article.

English language corrections in

1. Line 44, 45

2. Line 48,49 50, 51, 52, 53, 54 - grammar/tense correction

3. In Sample size determination

4. In Inclusion and exclusion criteria

5. In the discussion section.

Reviewer 2 Report

INTRODUCTION

Line 52 - the reference is missing

Line 55 - remove the point after (ref)

MATERIALS AND METHODS

Line 73 – Authors must separate the factors and put the patients ages in the inclusion factors

Exclusion factors should not be the opposite of inclusion factors. Thus, the authors must remove those that present as contrary to what they presented in the inclusion factors

DISCUSSION

“…Facial…” = “…facial…” – line 221

“…Our aim …” = “…the study aim…” – line 232

“…In our study…” = “…in this study…” – line 252

“…we have 252 found a lesser thickness…”= “…was found…”( Authors must correct the verbal form throughout the text) – line 253

Reviewer 3 Report

The paper "A CBCT study of labial alveolar bone thickness in the maxillary anterior region of a teaching hospital population in the Eastern province of Saudi Arabia" is of a little interest. The authors performed a well-documented study on bone thickness in the maxillary anterior region on 186 CBCT on 506 patients. The reason for the CBCT performed isn't clear (the authors don't explain the real reason, but ethical reasons advise against performing it on healthy patients). This study evaluates the bone thickness in the maxillary region in a purely descriptive way. It is not clear what the purpose of the study might be: the clinician is generally interested in evaluating the implant site if it's edentulous; and in any case it's not possible to give indications on the implant site with the teeth in place which is subject to many variables and cannot be predicted with such a study. In my opinion, this study does not bring any novelty in the scientific field, so it should be rejected.

Reviewer 4 Report

This is a nice scientific protocol, but please find here some remarks and recommandations of this reviewer  :

- why did the authors have a group of patients < 30 years old ? They found no difference in labial cortical bone thickness depending on the two age groups, but this seems not to believe because this kind of correlation is well known. Their discussion should more be focused on this point

- where all the examinated corridors dentate ? this is not mentioned in the manuscript, but it should...

- what do the authors mean by "periodontal health"  : this should be better detailed in the inclusion or exclusion criteria (severe periodontal disease is mentioned as exclusion criteria..does it mean that other periodontal diseased teeth are accepted for the study ?)

- what do the authors mean by "tooth inclinaison" ? The authors mention that the labial cortical bone thickness is depending on tooth inclinaison, but what about this inclinaison in their study (they mention also in the inclusion criteria that not any of the measured patients had an anormal tooth inclinaison..): the criteria about tooth inclinaison should be better detailed

- the reference n° 9 is the same as the reference n° 11 : this must be changed

- L219 : the legend for the colored columns fails

Round 2

Reviewer 3 Report

Although the authors have modified the paper for improvement, in my opinion this study doesn't bring any novelty in the scientific field so it should be rejected.